# Cardiolipin-Containing Lipid Membranes Attract the Bacterial Cell Division Protein DivIVA

**DOI:** 10.3390/ijms22158350

**Published:** 2021-08-03

**Authors:** Naďa Labajová, Natalia Baranova, Miroslav Jurásek, Robert Vácha, Martin Loose, Imrich Barák

**Affiliations:** 1Institute of Molecular Biology SAS, Dubravska Cesta 21, 845 51 Bratislava, Slovakia; 2Institute of Science and Technology Austria (IST Austria), Am Campus 1, 3400 Klosterneuburg, Austria; natalia.baranova@univie.ac.at (N.B.); martin.loose@ist.ac.at (M.L.); 3CEITEC and Faculty of Science, Masaryk University, Kamenice 5, 625 00 Brno, Czech Republic; jurasek.m@gmail.com

**Keywords:** *Clostridioides difficile*, DivIVA, lipid membrane, cardiolipin, phosphatidylglycerol

## Abstract

DivIVA is a protein initially identified as a spatial regulator of cell division in the model organism *Bacillus subtilis*, but its homologues are present in many other Gram-positive bacteria, including *Clostridia* species. Besides its role as topological regulator of the Min system during bacterial cell division, DivIVA is involved in chromosome segregation during sporulation, genetic competence, and cell wall synthesis. DivIVA localizes to regions of high membrane curvature, such as the cell poles and cell division site, where it recruits distinct binding partners. Previously, it was suggested that negative curvature sensing is the main mechanism by which DivIVA binds to these specific regions. Here, we show that *Clostridioides difficile* DivIVA binds preferably to membranes containing negatively charged phospholipids, especially cardiolipin. Strikingly, we observed that upon binding, DivIVA modifies the lipid distribution and induces changes to lipid bilayers containing cardiolipin. Our observations indicate that DivIVA might play a more complex and so far unknown active role during the formation of the cell division septal membrane.

## 1. Introduction

*Clostridioides difficile* is a Gram-positive obligate anaerobic bacterium and a primary cause of antibiotic-associated diarrhoea, and a frequent cause of nosocomial infections [1,2,3]. Its ability to produce toxins and to form spores makes it a highly aggressive and resistant infectious agent. In severe cases, it can even cause death of the infected patients. Worldwide, *C. difficile* infection rates and related deaths are comparable or even higher than infections and deaths caused by methicillin-resistant *Staphylococcus aureus* (MRSA) [2,3,4,5,6]. Hence, *C. difficile* infections cause serious problems and put extensive economic pressure on healthcare systems. The details of the *C. difficile* cell division and sporulation mechanisms and the functioning of the proteins involved in them are of great importance and could allow the development of drugs for disease treatment and preventing its spread.

One of the proteins that plays an important role during both vegetative cell division and asymmetric cell division during sporulation is DivIVA. DivIVA homologues are usually present in Gram-positive bacteria and have been studied for decades, especially in the model organism *Bacillus subtilis*. DivIVA is a late division protein and appears at the division site slightly later than the early division protein complex, which is led by the key division protein FtsZ [7,8,9]. DivIVA is localized not only at division site but during most of the cell cycle also at the cell poles, where it serves as a topological specifier for numerous proteins [10,11]. Importantly, DivIVA determines the subcellular position of the Min system (MinJ, MinD, and MinC), which is a negative regulator of FtsZ polymerization [12,13,14,15,16]. The Min system effectively blocks the formation of malignant polar septa at the cell poles and of additional division septa at areas close to the nascent division septum [12,17,18]. In *B. subtilis*, DivIVA is involved not only in cell division during vegetative growth but also in other important cellular processes, including competence or chromosome segregation during sporulation [10,16,19,20,21]. It is crucial for proper localization of RacA during sporulation [22,23], and Maf and ComN for competence [21,24]. Although *B. subtilis* DivIVA is not considered essential for vegetative growth, DivIVA mutant strains are filamentous [10,19], and unable to sporulate efficiently [10,19,25,26]. In other organisms, like *Streptomyces coelicolor*, *Mycobacterium tuberculosis*, *Mycobacterium smegmatis*, and *Corynebacterium glutamicum*, DivIVA homologues are essential and possess diverse functions [16,27,28,29]. For instance, in *Streptomyces coelicolor*, DivIVA is crucial for apical growth and proper morphogenesis [30].

DivIVA protein is a component of the Min system also in *C. difficile*, along with MinC, MinD, and MinE homologues. The presence of MinE likely enables oscillation of the *C. difficile* Min system, a dynamic repositioning of MinCDE proteins from one pole to other [31]. The oscillation of the Min system has been characterized in *Escherichia coli* and creates a gradient of Min protein complex, with its overall highest concentration at the cell poles and lowest in the middle of the cell [32]. Interestingly, the *E. coli* Min system can oscillate when replanted into *B. subtilis* cells [33]. Similarly, oscillation of *C. difficile* MinD and MinE was observed when expressed heterogeneously in *B. subtilis* cells [31]. The coinciding presence of DivIVA, and its capability to interact with MinD and to stably localize to division site and persist at the cell poles [34] possibly influences the Min oscillation in *C. difficile*. Whether and how DivIVA affects the *C. difficile* Min system dynamics are so far unknown.

The key aspect of DivIVA in all the diverse processes, in which it is involved, is its correct localization. As mentioned above, in *B. subtilis*, DivIVA is recruited to the division site during the later stages of cell division [7]. Rather than protein–protein interactions, however, a negative membrane curvature was suggested to be the key factor determining the recruitment of DivIVA at the division site [35,36]. Negative membrane curvature is found at the cell division site because of the membrane invagination, and also at the cell poles; consequently, these are the sites where DivIVA is usually present [10,11]. Although membrane curvature was suggested to be the crucial factor attracting DivIVA to the division site, recent computer simulations suggested that lipid-specific interactions might be involved in its localization. The N-terminal domain of DivIVA from *S. coelicolor* exhibited an elevated affinity for anionic phospholipids, particularly cardiolipin [37]. However, this preference has not been experimentally confirmed until now.

In this work, we focused on DivIVA from *C. difficile*, about which only limited information is available. Our results indicate that DivIVA not only has a higher affinity for anionic phospholipids, especially cardiolipin, but that it is also capable of remodelling the lipid bilayer. Its binding to cardiolipin-containing lipid bilayers probably enhances cardiolipin clustering and induces bilayer deformation. Our findings indicate that DivIVA might not only attract crucial division proteins to the division plane but might play an active role in the shaping or scaffolding of the developing septal membrane during cell division.

## 2. Results

### 2.1. DivIVA_Cd_ Binds More Efficiently to Cardiolipin-Containing Membranes In Vitro

The structural similarities between DivIVA from *B. subtilis* (DivIVA_Bs_) and *C. difficile* (DivIVA_Cd_) suggest that DivIVA_Cd_, like DivIVA_Bs_, can bind to the membrane [35,38]. It is known that the N-terminal domain of DivIVA_Bs_ is involved in its attachment to the membrane [35,36,38]. It forms a coiled-coil dimer with crossed loops, which expose a combination of a hydrophobic and one or more positively charged amino acid residues on its surface. The hydrophobic residue is thought to be inserted below the lipid head group and interact with the phospholipid tails, while the positively charged residue(s) (usually arginine) interact with the lipid phosphate groups [37]. These residues are conserved across several species [38]. For instance, DivIVA_Bs_ contains phenylalanine and arginine, while DivIVA_Cd_ has leucine and arginine (Appendix A).

To examine the ability of DivIVA_Cd_ to bind to lipids, we performed sedimentation assays with DivIVA_Cd_ and small unilamellar vesicles (SUVs) (Figure 1A). We tested SUVs composed of 70% phosphatidylcholine (PC) and 30% phosphatidylglycerol (PG) (L1: 70% PC and 30% PG), a lipid mixture commonly used to mimic bacterial membranes [39,40]. The membrane of *C. difficile* contains, besides PG (23–35%), a significant amount (12–20%) of cardiolipin (CL), another negatively charged phospholipid [41]. We therefore studied vesicles containing 5% (L4: 65% PC, 30% PG, and 5% CL) and 15% cardiolipin (L6: 55% PC, 30% PG, and 15% CL). We also examined SUVs composed of 100% PC (L2) to see if DivIVA_Cd_ binds to vesicles composed of only neutral lipids.

In the negative control without SUVs and with L2, similar amounts of DivIVA_Cd_ (95 ± 5% and 95 ± 3%, respectively) remained in the soluble fraction (Figure 1B,C). This indicates that DivIVA_Cd_ is not binding to L2 vesicles in the sedimentation assay. With L1 SUVs, slightly less DivIVA_Cd_ was in the soluble fraction (90 ± 5%), which points toward weak binding. Interestingly, cardiolipin-containing vesicles enhanced DivIVA_Cd_ sedimentation. In the presence of 5% cardiolipin (L4), 81 ± 5% of the DivIVA_Cd_ appeared in the soluble fraction and 19% of the DivIVA_Cd_ was in the pellet (Figure 1B,C). The effect was even more pronounced in the presence of 15% cardiolipin (L6): 52 ± 7% of the DivIVA_Cd_ was soluble and 48% of the DivIVA_Cd_ bound to SUVs (Figure 1B,C). The differences between the binding of DivIVA_Cd_ to the L1, L4, and L6 vesicles are all statistically significant (Appendix A), indicating that DivIVA_Cd_ has a preference for cardiolipin.

To test whether the N-terminal domain of DivIVA_Cd_ is essential for its binding to the membrane, we prepared an N-terminally truncated DivIVA_Cd_ protein, lacking the first 60 amino acids (Δ60-DivIVA_Cd_). We could not detect statistically significant differences between the amount of Δ60-DivIVA_Cd_ in the pellet without SUVs and in the presence of the L1, L2, L4, and L6 lipid mixtures (Figure 1B,C, Appendix A). Without its N-terminal domain, DivIVA_Cd_ lost its ability to bind even to cardiolipin-containing vesicles, confirming that the N-terminal domain is involved in DivIVA_Cd_ membrane binding.

### 2.2. DivIVA_Cd_ Binds to a Supported Lipid Bilayer

To confirm the preferential binding of DivIVA_Cd_ to cardiolipin-containing membranes, we employed a quartz crystal microbalance with dissipation (QCM-D). QCM-D is an acoustic method, which monitors changes in frequency and dissipation after the binding of a macromolecule to a defined substrate layer on a QCM-D sensor. The adsorption of DivIVA_Cd_ to a supported lipid bilayer should produce negative frequency (Δf) and positive dissipation (ΔD) shifts (Figure 2A). The changes in frequency and dissipation depend on both the adsorbed mass and the mechanical properties and morphological features of the biomolecular film [42,43,44].

Using this method, we analysed how DivIVA_Cd_ lipid binding depends on the concentrations of both cardiolipin and DivIVA_Cd_. We tested mixtures containing either no CL (L1) or 2.5% CL (L3), 5% CL (L4), and 7.5% CL (L5). Unfortunately, formation of a homogenous bilayer on the sensor chip using lipid mixtures with CL concentrations above 7.5% was not possible, so we could not test the 15% CL composition (L6) used in the sedimentation assay. We note that other researchers have encountered similar difficulties forming a contiguous SLB on sensor chips with higher cardiolipin concentrations [45].

After forming the lipid bilayers from lipid mixtures L1, L3, L4, or L5, washing with buffer, and obtaining a stable base line, DivIVA_Cd_ was added and its concentration stepwise increased from 0.25 µM to 3.0 µM. On SLBs consisting of 70% PC and 30% PG (L1), the addition of DivIVA_Cd_ produced the highest Δf response of −18.2 ± 1.3 Hz with 3.0 µM DivIVA_Cd_ at overtones 3, 5, 7, and 9 (Figure 2B). The presence of 2.5%, 5%, and 7.5% CL (L3, L4, and L5) in the lipid mixtures enhanced the binding of DivIVA_Cd_ (Figure 2C–E). At 3.0 µM DivIVA_Cd_, the maximal Δf was −37.0 ± 3.4 Hz for 2.5% CL, −53.3 ± 3.2 Hz for 5% CL, and −71.0 ± 5.1 Hz for 7.5% CL. The preference of DivIVA_Cd_ for cardiolipin-containing SLBs is visible from a summary plot showing the maximum frequency changes with increasing protein concentration (Figure 2F). The binding is non-linear and the Hill equation can be fitted well to the data. Without cardiolipin, the Hill coefficient is 0.4, with 2.5% CL 0.7, 5% CL 1.0, and with 7.5% CL 1.6, which suggests that DivIVA_Cd_ binding is enhanced at higher CL concentrations and is positively cooperative. K_D_ was calculated to be in the micromolar range (5.7 µM at 2.5% CL, 1.3 µM at 5% CL, and 0.8 µM at 7.5% CL). We also tested the N-terminal truncated Δ60-DivIVA_Cd_ construct and measured its binding to the L1 and L4 lipid mixtures (Appendix A). The addition of Δ60-DivIVA_Cd_ to these SLBs resulted in a negligible Δf response even after a protein concentration increase up to 2.0 µM (Appendix A).

The advantage of the QCM-D measurements is that it allows estimation of the viscoelastic properties of the biofilm, i.e., lipid membrane and adsorbed protein, according to the behaviour of different overtones [44,46,47]. In the presence of an SLB that does not contain CL (L1), the ΔD between different overtones overlap after DivIVA_Cd_ addition (Figure 2B and Appendix A), which suggests that the biofilm is rigid and homogeneous. At 2.5% CL (L3, Figure 2C and Appendix A), 5% CL (L4, Figure 2D and Appendix A), and 7.5% CL (L5, Figure 2E and Appendix A), the different overtones do not overlap and are nonlinear, which means that the biofilm is soft and is not homogenous. After buffer wash and a partial DivIVA_Cd_ detachment, the dissipation shift spread between different overtones persists. Possibly the film remains soft, and DivIVA_Cd_ forms stable complexes on the bilayer or it causes irreversible changes to the morphology of the bilayer, or both. Because of the softness and heterogeneity of the biofilm, we did not attempt to estimate the mass and thickness of the adhering DivIVA_Cd_.

In summary, our measurements demonstrated that DivIVA_Cd_ is capable of binding to liposomes and to SLBs. Its binding is strongly dependent on the lipid composition and cardiolipin enhances it. Low but measurable binding of DivIVA_Cd_ to 30% PG-containing membranes (L1), especially in the QCM-D assay, indicates that the plausible molecular explanation for DivIVA_Cd_ binding is an electrostatic interaction. PG is a negatively charged phospholipid with a net charge of −1 at pH 7.4, while CL, or diphosphatidylglycerol, bears a −2 net charge at the same pH. The N-terminal domains of DivIVA_Cd_ and other DivIVA homologues contain a number of positively charged amino acids, especially around the N-terminal loop responsible for the membrane interaction. An electrostatic interaction between the positively charged N-terminal DivIVA domain and the negatively charged phospholipids-containing membranes has previously been suggested as the mechanism of DivIVA–membrane contact [37,38]. Interestingly, the QCM-D results suggest that DivIVA_Cd_ induces morphological changes only to SLBs containing cardiolipin. We also confirmed that the N-terminal domain is essential for the DivIVA_Cd_–lipid interaction.

### 2.3. DivIVA_Cd_ Deforms the Supported Lipid Bilayer

The QCM-D measurements of DivIVA_Cd_ binding to CL-containing SLBs indicated that DivIVA_Cd_ binding might introduce morphological changes to the membrane. To determine the possible changes to the SLB, we examined SLBs with fluorescently labelled lipids using TIRF (total internal reflection fluorescence) microscopy. We analysed the distribution of fluorescently labelled cardiolipin (TopFluor^®^ Cardiolipin, TF-CL) before and after DivIVA_Cd_ addition to individual SLBs. Unfortunately, already with SLBs, in which the only cardiolipin present was the 0.1% TopFluor^®^ Cardiolipin (L1*: 69.9% PC, 30% PG and 0.1% TF-CL), the fluorescent signal was not uniformly distributed before DivIVA_Cd_ addition (Figure 3A). To some extent, TopFluor^®^ Cardiolipin formed clusters by itself. After DivIVA_Cd_ addition, these foci became larger and brighter (Figure 3B) clearly, as when DivIVA_Cd_ binds to this SLB, the TopFluor^®^ Cardiolipin clustering is enhanced. In the presence of 5% CL (L4*: 64.9% PC, 30% PG + 5% CL, and 0.1% TF-CL), a larger number of small fluorescent foci were present before DivIVA_Cd_ addition (Figure 3C), and, after the addition of DivIVA_Cd_, even larger clusters than in the sample without cardiolipin became visible (compare Figure 3B,D). To examine the dynamics of the lipids within the SLB, we also performed FRAP experiments both before and after DivIVA_Cd_ addition. We observed fluorescence recovery in all experiments, indicating that the CL clusters are not static assemblies and that DivIVA_Cd_ binding does not prohibit the exchange of lipids within the clusters (Appendix A).

To avoid the clustering of TopFluor^®^ Cardiolipin, which possibly masks the effect of DivIVA_Cd_ on such SLBs, we used another fluorescently labelled lipid, lissamine rhodamine phosphatidylethanolamine (Rh-PE). Phosphatidylethanolamine (PE) is a zwitterion whose head contains a negatively charged phosphate and a positively charged free amine, and also the fluorescent moiety is uncharged at neutral pH [48], making it unlikely that DivIVA_Cd_ would be attracted by it. On the other hand, as PE’s overall shape is conical, it will be abundant in areas with negative curvature [49]. Thus, if the interaction of DivIVA_Cd_ with CL leads to curvature formation, we should also expect to observe the redistribution of Rh-PE fluorescence within the SLBs.

Without DivIVA_Cd_ the Rh-PE fluorescent signal was mostly uniform in the SLB; only very faint foci were visible either with or without CL (Figure 4A,C). With SLBs that did not contain CL (L1**: 69.9% PC, 30%PG, and 0.1% Rh-PE), the fluorescence distribution after DivIVA_Cd_ addition remained unchanged (Figure 4B). In an SLB containing 5% CL (L4**: 64.9% PC, 30% PG, 5% CL, and 0.1% Rh-PE), the addition of DivIVA_Cd_ affected the fluorescence signal distribution substantially (Figure 4D,E). Besides larger round clusters, we also observed a variety of different patterns, reminiscent of membrane vesicles or protrusions (Figure 4E).

The TIRF microscopy results show that DivIVA_Cd_ rearranges SLBs containing cardiolipin. Most likely, the binding of DivIVA_Cd_ to the lipid bilayer causes changes in the lipid distribution, the formation of CL clusters, and possibly bilayer deformations. We did not observe changes in the lipid distribution in bilayers without cardiolipin (L1), even though our previous QCM-D data indicate that DivIVA_Cd_ does bind to them.

### 2.4. DivIVA Also Shows a Higher Affinity for Cardiolipin-Containing Membranes in Molecular Dynamics Simulations

To investigate the interaction between the N-terminal domain of DivIVA_Cd_ and different lipids at the molecular level, we performed molecular dynamics simulations. First, we prepared a homology model of the N-terminal domain of *C. difficile* DivIVA_Cd_ using the DivIVA_Bs_ N-terminal domain as a template. This modelling indicated that the DivIVA_Cd_ N-terminal domain has a similar charge distribution as the corresponding domain of *B. subtilis* DivIVA (Figure 5A). We tested three membrane compositions in the molecular dynamics simulations: 100% PC (L2), 5% CL (L4), and 15% CL (L6). We found that the N-terminal domain interacts with all three membranes (Figure 5B–D), although in one out of the two simulations of the L2 membrane, the domain desorbed from the membrane. This unbinding indicates that the N-terminal domain has a lower affinity for the 100% PC membrane. In the two membranes with 5% and 15% cardiolipin (L4 and L6), the domain remained bound throughout the whole 200 ns simulation, indicating a stable interaction on this time scale. When bound, the domain remained perpendicular to the membrane (Figure 5). Only the tip of the domain (residues 14–18) was inserted into the membrane where two arginine residues interacted with the phosphates of the head groups and two leucine residues were inserted into the hydrophobic core. We also observed that cardiolipin formed clusters (areas of increased average density) around the domain tip, particularly in the membrane with the higher cardiolipin content (L6). In the membrane with the lower cardiolipin content (L4), the clusters were significantly smaller. The size of the clusters might be affected by the length of our simulations, since a lower membrane cardiolipin content decreases the opportunity for clusters to form. In simulations with both 5% and 15% cardiolipin (L4 and L6), a tiny increase in the density of phosphatidylglycerol was also observed. Taken together, our simulations suggest that the N-terminal domain has a higher affinity for anionic lipids, particularly cardiolipin, and are in agreement with the in vitro experiments described above.

## 3. Discussion

Although bacterial cells lack membrane compartmentalisation similar to eukaryotic organelles, bacterial proteins and protein complexes are substantially organized within the cell and many localize to specific sites with high precision. To understand the complexity of the most essential cell processes (e.g., cell division), it is crucial to study the mechanisms that determine this precision. For example, numerous membrane-binding proteins, including the DivIVA division proteins, show site-specific binding in vivo [10,11]. How these proteins find their specific destinations on the cell membrane, especially the cell poles and the division site, is a question of great interest. Membrane curvature sensing has been proposed to be one possible mechanism used by proteins to localize to specific subcellular positions [50]. It was shown previously that *B. subtilis* DivIVA can recognize concave (negative) membrane curvature [35,36].

DivIVA homologues can be found in numerous members of the Gram-positive bacterial phylum Firmicutes (e.g., *Bacillus* sp., *Clostridioides* sp.) and Actinobacteria (e.g., *Streptomyces* sp., *Mycobacteria* sp.), and they perform diverse functions. The complexity of the intracellular environment and the limited research tools complicate the study of the mechanisms responsible for their proper positioning and functioning in vivo, however. In this work, we characterized in vitro DivIVA from the human pathogen *C. difficile* (DivIVA_Cd_), which has a high degree of similarity to *B. subtilis* DivIVA (DivIVA_Bs_). We provide the first detailed study on the interaction between DivIVA_Cd_ and lipids combined with molecular dynamics simulations. We show that DivIVA_Cd_ binds to SUVs containing cardiolipin more efficiently than to SUVs without cardiolipin. We observed that cardiolipin attracts DivIVA_Cd_ to lipid membranes when formed into the supported lipid bilayers (SLBs) in QCM-D assays.

We also examined the involvement of the DivIVA_Cd_ N-terminal domain in binding to lipids. The 3-D structure of DivIVA_Cd_ is not known, but our homology model of the N-terminal domain of DivIVA_Cd_ shows it is probably similar in structure and surface charge distribution to the known DivIVA N-terminal domains. Molecular dynamics simulations of its binding to lipid bilayers revealed that the N-terminal domain has an affinity for anionic lipids, particularly cardiolipin. This is in agreement with our in vitro experimental results on full-length and N-terminally truncated DivIVA_Cd_. In contrast to the strong binding of the full-length protein, we did not detect any binding between the N-terminally truncated DivIVA_Cd_ and any SUVs in sedimentation assays, nor to any SLB in QCM-D experiments, including those with cardiolipin.

The preferential binding of DivIVA_Cd_ to cardiolipin identified here is also consistent with the available in vivo data from the *B. subtilis* DivIVA homologue. DivIVA_Bs_ initially localizes to the developing division septum in vivo, from which the future pole arises, and remains at the cell poles [10,11]. When the septal membrane invaginates during division septum formation, negative curvature forms. Due to its conical shape, cardiolipin accumulates in the parts of the membrane with higher curvature [51], and the accumulation of cardiolipin necessarily enhances curvature formation. Interestingly, the major cardiolipin synthase ClsA also localizes to the membrane at the division site, which suggests that additional cardiolipin is synthesized at the division site [52]. Cardiolipin is probably also abundant at the at cell poles, which have an inner negative curvature. In both *E. coli* and *B. subtilis*, cardiolipin-rich polar domains have been identified by staining the cell with the hydrophobic fluorescent dye 10-*N*-nonyl acridine orange (NAO) [53,54]. Though the use of NAO as a specific dye for visualizing cardiolipin domains has recently been called into question [55], the existence of polar cardiolipin-rich domains has not yet been ruled out, even though their existence at the cell poles is only indirectly indicated. For example, the membranes of minicells, DNA-free cells that form after anomalous polar cell division, are enriched in cardiolipin [56]. In addition, the membranes of spores, differentiated cell types arising from a special type of polar cell division, have a higher cardiolipin content [57]. It is highly probable that if cardiolipin-rich domains exist, they will be present throughout the bacterial kingdom and could possibly serve as recognition and anchoring areas for proteins of different functions, possibly including DivIVA.

According to our results, DivIVA_Cd_ not only binds to cardiolipin-containing membranes, but it also possibly changes the morphology of these membranes. Using EM, Lenarcic and co-workers [35] showed previously that the morphology of vesicles was also impaired upon DivIVA_Bs_ binding. These findings, together with the presence of DivIVA at the developing division septum site, indicate that DivIVA might have a more important role during cell division than previously thought. It is possible that DivIVA plays an active role in septal membrane formation or may serve as a scaffold during membrane invagination. Some in vivo data are consistent with this hypothesis. Specifically, DivIVA_Bs_ appears at the division site slightly sooner than the septal membrane [7]. In cultures of synchronised germinating spores, more cells have a DivIVA_Bs_-GFP signal at the mid-cell than a visible septum. Similarly, deconvolution fluorescence microscopic imaging of *B. subtilis* cells showed that part of the cells formed a visible DivIVA_Bs_ ring before the septal membrane develops, suggesting that DivIVA_Bs_ localizes to septa at the onset of membrane invagination [11].

Based on our findings, we propose a modified model for the mechanism of specific DivIVA localization. Initially, DivIVA binds preferentially to parts of the membrane where cardiolipin or a locally higher concentration of cardiolipin is present (Figure 6). Upon binding, DivIVA possibly attracts more cardiolipin to these areas and consequently more DivIVA molecules bind. DivIVA binding or cardiolipin accumulation causes re-modelling of the bilayer, leading to the migration of even more cardiolipin to these positions. This positive-feedback mechanism enhances curvature formation, which in turn attracts more DivIVA. Possibly both DivIVA’s affinity for cardiolipin and its affinity for negative curvature act together and are responsible for DivIVA’s specific cellular localization. DivIVA’s membrane remodelling capability and whether it plays some role during the cell division requires further research.

## 4. Materials and Methods

### 4.1. General Methods

For cloning, the *E. coli* MM294 or *E. coli* DH5 strains were used and transformations were performed following standard protocols [58,59]. The strains were grown in LB medium [60] supplemented with 50 µg/mL kanamycin at 37 °C. A complete list of all *E. coli* strains and plasmids used in this study is given in Appendix A; oligonucleotide primers are given in Appendix A. PCR fragments were amplified from the chromosomal DNA of the *C. difficile* 630 strain (a kind gift from Prof. Neil Fairweather, Imperial College London). DivIVA_Cd_ was expressed without tag or as His-SUMO fusion protein. To construct pET26-DivIVA_Cd_ (DivIVA_Cd_ without tag), the PCR-amplified DivIVA_Cd_ gene, digested with NdeI/BamHI, was ligated into a pET-26b (Novagen) plasmid cleaved with the same restriction enzymes. The N-terminaly truncated Δ60-DivIVA_Cd_ was prepared in a similar way: a PCR-amplified DivIVA_Cd_ gene lacking its first 180 bp, digested with NdeI/BamHI, was ligated into a NdeI/BamHI cleaved pET-26b (Novagen) plasmid, giving rise to plasmid pET-26b-Δ60-DivIVA_Cd_ (Appendix A). To construct pTB-DivIVA_Cd_ (His-SUMO-DivIVA_Cd_), the DivIVA_Cd_ gene was amplified with PCR, cleaved with SapI/BamHI, and ligated into pTB146 [61]. Cloning into SapI site enables expression of fusion protein, which after Ulp1 protease cleavage will not have additional amino acids in comparison to the native protein.

### 4.2. DivIVA_Cd_ and Δ60-DivIVA_Cd_ Protein Expression and Purification

We used a slightly modified DivIVA_Bs_ protein purification protocol [62] for both full-length DivIVA_Cd_ and its truncated version, Δ60-DivIVA_Cd_. For DivIVA_Cd_ and Δ60-DivIVA_Cd_ overexpression, *E. coli* BL21(DE3)pLysS cells containing pET26-DivIVA_Cd_ or pET26-Δ60-DivIVA_Cd_ were grown at 37 °C in LB medium and 0.5 mM IPTG was added at OD_600_ ∼0.7. After 3 h of induction at 37 °C, cells were harvested and stored at −80 °C until processed. The cell pellet was thawed and resuspended in buffer A (50 mM Tris·HCl, pH 8.0, 100 mM NaCl, 1 mM EDTA) plus cOmplete™ Protease Inhibitor Cocktail tablets (Roche) and sonicated to lyse the cells. The lysates were centrifuged at 85,000× *g* at 6 °C for 30 min. DivIVA is a negatively charged protein and forms higher order structures; thus, anion-exchange followed by size exclusion chromatography is used for purification. The supernatants were applied to a HiPrep Q FF 16/10 column (GE Healthcare), washed with 3 column volumes (CV) of buffer A, and eluted using 3 CV of a linear gradient with up to 100% buffer B (50 mM Tris·HCl, pH 8.0, 1 M NaCl, 1 mM EDTA). DivIVA_Cd_ was usually found in the 700 mM NaCl fractions. Selected fractions were pooled, concentrated and loaded onto a Sephacryl 16/60 S-300 HR column (GE Healthcare), and eluted with a buffer composed of 50 mM Tris·HCl, pH 8.0, and 100 mM NaCl. Most of the DivIVA_Cd_ eluted in the void volume and in the first third of the column volume, indicating that DivIVA_Cd_ forms large polymers. The best fractions were pooled, concentrated if necessary, aliquoted, and stored at −80 °C. The N-terminally truncated form of DivIVA_Cd_ (Δ60-DivIVA_Cd_) was purified in a similar way.

Alternatively, we expressed DivIVA_Cd_ in fusion with His-SUMO-tag, which can be cleaved by His-Ulp1 protease. *E. coli* BL21(DE3)pLysS cells containing pTB-DivIVA_Cd_ were grown at 37 °C in LB medium and 0.5 mM IPTG was added at OD_600_ ∼0.6. After 1 h of induction at 30 °C, the temperature was lowered to 18 °C and culture was grown overnight. Cells were harvested and stored at −80 °C until processed. The cell pellet was thawed and resuspended in buffer A (50 mM Tris·HCl, pH 7.5, 150 mM NaCl, 50 mM imidazole) plus cOmplete™ Protease Inhibitor Cocktail tablets (Roche) and sonicated to lyse the cells. The lysates were centrifuged at 85,000× *g* at 6 °C for 30 min. The supernatant was loaded to IMAC Ni-NTA column, extensively washed, and eluted using a 50 mM–500 mM imidazole linear gradient. The chosen aliquots were pooled and cleaved with in-house prepared His_6_-Ulp1 protease overnight at 10 °C. Using PD-10 desalting columns the buffer was changed back to 50 mM Tris·HCl, pH 7.5, 150 mM NaCl, 50 mM imidazole and samples were re-loaded to the Ni-NTA column. The flow-through fractions contained DivIVA_Cd_ while His_6_-SUMO and His_6_-Ulp1 protease bound to the column. Fractions containing DivIVA_Cd_ were pooled, and the buffer was changed to 50 mM Tris·HCl, pH 8.0, and 100 mM NaCl on PD-10 desalting columns. If necessary, the protein was concentrated, aliquoted, and stored at −80 °C. This approach yielded purer and more soluble protein. Importantly, the SEC profile or the preference for CL was comparable to DivIVA_Cd_ expressed and purified without the tag.

### 4.3. Preparation of Small Unilamellar Vesicles (SUVs)

Different lipid mixtures were used in the assays and were prepared by mixing the synthetic lipids DOPC (1,2-dioleoyl-sn-glycero-3-phosphocholine), DOPG (1,2-dioleoyl-sn-glycero-3-phospho-rac-1-glycerol), and TOCL (1,3-bis[1,2-dioleoyl-sn-glycero-3-phospho]-glycerol), all from Avanti Polar Lipids, Inc. Lipids were solubilised in chloroform and mixed to obtain 5 mM stock solutions as listed in Appendix A. The chloroform was evaporated under a stream of nitrogen and the lipids were further vacuum dried for one hour at laboratory temperature. To prepare SUVs for SLB formation, the dried lipid mixtures were hydrated in a buffer containing 25 mM Tris·HCl, pH 7.5, 50 mM KCl, frozen in liquid N_2_ and thawed 5 times, and tip sonicated with cooling until the solution became translucent. To prepare SUVs for sedimentation assays the dried lipid mixtures were dissolved in a buffer containing 25 mM Tris·HCl, pH 7.5, 50 mM KCl, and 30% sucrose and the same protocol was followed as above for SUVs without sucrose.

To visualise the lipid bilayer, fluorescent derivatives of phosphatidylethanolamine ((1,2-dioleoyl-sn-glycero-3-phosphoethanolamine-N-(lissamine rhodamine B sulfonyl), Avanti Polar Lipids); Rh-PE) or cardiolipin ((1,1′,2,2′-tetraoleoyl cardiolipin[4-(dipyrrometheneboron difluoride)butanoyl], Avanti Polar Lipids); TF-CL) were added to the lipid mixtures at 0.1% concentrations and the SUVs were prepared as usual. All fluorescent lipid mixtures are listed in Appendix A.

### 4.4. Sedimentation Assay

Prior to each experiment, protein samples were thawed on ice and centrifuged for 10 min at 16,000× *g* to remove potential protein aggregates. A reaction mixture usually contained 5 µM DivIVA_Cd_ or 10 µM Δ60-DivIVA_Cd_ and 1 mM SUVs and was incubated for 15 min at 30 °C. Samples were centrifuged for 10 min at 16,000× *g* and 30 °C. Even this low centrifugation force was sufficient to sediment the lipid vesicles to which DivIVA_Cd_ was bound. We also tested higher centrifugation force (100,000× *g*) and obtained similar results (not shown). The supernatant containing the unbound protein fraction was collected and 4× SDS sample buffer was added to reach a 1× final concentration. Pellets containing the liposome-bound protein fraction were resuspended directly in 1× SDS sample buffer. Samples were analysed using SDS-PAGE and the ratio between bound and unbound protein was quantified using the Gel Analysis Fiji plugin. The results were plotted in GraphPad Prism 8 and evaluated using a two-tailed *t*-test. P values were not corrected for multiple comparisons and apply individually to each value, not to the entire family of comparisons.

### 4.5. QCM-D Measurements

To measure the binding of DivIVA_Cd_ to SLBs, a QCM-D (QSense Analyzer, Biolin Scientific) was used. The frequency (Δf) and dissipation (ΔD) changes can be obtained from QCM-D by measuring the piezoelectric properties of a quartz sensor upon biomolecule binding [63]. Prior to QCM-D measurements, the silica-coated QCM-D sensors were cleaned in 2% SDS aqueous solution for 15 min, washed extensively with ultrapure water, and blow-dried. All measurements were performed at 23 °C at 5 overtones, *n* = 3, 5, 7, 9, 11, corresponding to resonance frequencies of 15, 25, 35, 45, and 55 MHz, respectively. The flow cell was washed with ultrapure degassed water, and the solution was then exchanged for SLB buffer (50 mM Tris buffer, pH 7.4, 300 mM KCl) until a stable frequency signal was observed. To form the SLB, the 0.1 mM SUVs mixtures were injected onto the sensor chip in the presence of calcium ions (5 mM CaCl_2_) in the SLB buffer at a constant flow rate of 50 µL/min. The vesicles adsorb to the sensor surface and the calcium enhances their rupture and SLB formation [64,65]. The sensor chip was then washed with binding buffer (50 mM Tris buffer, pH 7.4, 250 mM KCl, 5 mM MgCl_2_) until a profile characteristic for SLB formation was acquired. Protein solutions containing increasing concentrations of DivIVA_Cd_ (0.25–3 µM) were added to the SLB surface at a constant flow rate of 30 µL/min. After loading the sample, the binding was measured until a stable base line was obtained. Finally, the flow cells containing the adsorbed DivIVA_Cd_ protein were rinsed with reaction buffer. Data were analysed and plotted using Microsoft Excel and GraphPad Prism 8.

### 4.6. TIRF Microscopy and Image Analysis

The sample chambers for TIRF microscopy were freshly prepared as published previously [40,66]. Briefly, the chambers were created by attaching a plastic ring made from a cut PCR reaction tube on an extensively cleaned glass cover slip using ultraviolet glue. The coverslips’ cleaning procedure involved washing with Piranha solution (a 3:1 mixture of sulfuric acid and 30% hydrogen peroxide) and plasma treatment (Diener Zepto plasma system). To form the supported lipid bilayer, 5 mM SUV stock solutions were first diluted in SLB buffer (50 mM Tris-HCl pH 7.4, 300 mM KCl) to obtain 1 mM lipid solutions and CaCl_2_ was added to a final concentration of 3 mM. Then, 30 µL of the lipid solution were added to each chamber and incubated for 1 h at laboratory temperature in the dark. Afterwards, 30 µL of SLB buffer were added to each chamber to obtain a 60 µL final volume. Chambers were washed 5× with 100 µL of SLB buffer to eliminate non-fused vesicles and 5× with 100 µL of binding buffer (50 mM Tris-HCl pH = 7.4, 150 mM KCl, 5 mM MgCl_2_) to exchange the buffer. For better illustration, the Jove protocol can be followed [40]. TIRF microscopy imaging was performed using a TIRF microscope iMic equipped with an image splitter (Andor Tucam) and two Ixon 897 cameras using 60× or 100× oil immersion objectives at 20 °C operated by LA (Lifeaquisition) software, version: 6.2.1. The laser power was adjusted depending on the signal intensity.

Image processing and analysis were carried out with the Fiji image processing package [67]. To reduce the noise in the images, the Anisotropic Diffusion 2D plugin was used with the default settings. To determine the area of the fluorescent signal, the Particle Analysis built-in function was employed, with MaxEntropy or Triangle threshold algorithms. In all cases, the same threshold settings were applied to compare images before and after DivIVA_Cd_ addition in the respective sample. The Microsoft Excel Histogram function was used to calculate the area distribution with a Bin range from 0 to 4 µm^2^, with 0.2 µm^2^ steps. The data were analysed and plotted in GraphPad Prism 8.

For the FRAP experiments shown in Appendix A, circular ROI with ~10 µm diameter was photo-bleached at full laser power using a 488 nm laser for TF-CL and 561 nm laser for Rh-PE with 2 ms/µm^2^ dwell time. The recovery of fluorescence was monitored at lower laser power (up to 30%) with a 50 ms exposure times and 5 s intervals. The changes in fluorescence intensities along the axis indicated in Appendix A were measured in Fiji and plotted in GraphPad Prism 8.

### 4.7. Homology Modelling

The homology model of the DivIVA *C. difficile* N-terminal domain was constructed using its *B. subtilis* homologue (PDB code 2WUJ) [38] as a template. The homology model itself was created by Swiss-Model [68]. The stability of the model was tested in a 100-ns-long simulation and its final configuration was used as the initial configuration for all simulations with membranes. The model was stable throughout the whole 100 ns trajectory, with its RMSD converging within the initial first nanoseconds and remaining stable until the end of the simulation.

### 4.8. Molecular Dynamics (MD) Simulations

All MD simulations were performed using GROMACS 5.1.2 [69], the CHARMM36 force-field [70], and the TIP3P water model [71]. All simulations were conducted at 309.15 K and 1 bar in a 150 mM aqueous solution of NaCl at neutral pH. Simulations were performed in the NPT ensemble with periodic boundary conditions. The system temperature was held constant with the Nosé-Hoover thermostat [72,73] with a time constant of 1 ps and the pressure was controlled using the Parrinello-Rahman barostat [74] with a coupling time of 5 ps and a compressibility of 4.5 × 10^−5^ bar^−1^. The solute and solvent were coupled to separate thermostats. Isotropic pressure coupling was applied in simulations with fully solvated protein. In systems with membranes, semi-isotropic coupling was used with the pressure normal to the membrane within the membrane plane coupled separately. Centre of mass movement was removed separately for the solvent and membrane with protein every 5 ps. Long-range electrostatics were modelled with PME [75,76] with a direct space cut-off of 1.2 nm. Short-range van der Walls interactions were cut-off at 1.2 nm with a force-shift applied from 1.0 nm. All hydrogen atoms bound to heavy atoms were constrained with LINCS [77]. The equations of motion were integrated using a leap-frog integrator with a Verlet neighbour search using a 0.005 kJ mol^−1^ps^−1^ buffer.

#### 4.8.1. System Preparation for MD Simulations

Membrane configurations were prepared with the CHARMM-GUI [78,79,80,81]. All membranes were solvated in a 150 mM aqueous solution of NaCl with additional ions to neutralize the system. Membranes were equilibrated until the area-per-lipid (APL) converged to steady state (roughly 50 ns). We investigated the interactions between the DivIVA_Cd_ N-terminal domain and lipid membranes containing dioleoyl phosphatidylcholine (DOPC), dioleoyl phosphatidylglycerol (DOPG), and tetraoleylcardiolipin with a −2 e charge (TOCL2). Three different lipid compositions were considered: pure DOPC, a DOPC-DOPG-TOCL2 mixture with a mol ratio of 13:6:1, and a second mixture with a ratio of 11:4:3. These correspond to 65% DOPC:30% DOPG:5% TOCL and 55% DOPC:30% DOPG:15% TOCL, respectively. Each system consisted of 320 lipids equally distributed in each leaflet. The initial configurations of the membrane-protein simulations were generated by adding the homology model of the N-terminal domain to the proximity of the equilibrated membrane with no solvent. The N-terminal domain was positioned with the domain tip (residues 14–18) facing the membrane. This orientation was chosen based on the reported preference of these loop residues for interacting with the membrane [38]. This membrane and domain system was subsequently solvated with an explicit water solution of 150 mM NaCl. The excess charge of the system was neutralized with additional ions. The systems were then equilibrated for 100 ns followed by 200-ns-long simulations during which the protein–membrane interactions were sampled and evaluated. Each membrane composition was simulated in two replicas, which differed in the initial membrane configuration to obtain independent sampling [82].

#### 4.8.2. The Domain Orientation

The orientation of the domain with respect to the membrane was calculated using the orthonormal vectors within the domain. We used the same definition as previously [37] for consistency and easier comparison. The first vector was aligned with the long axis of the domain and was defined by the centre of mass of the C_α_ atoms of two Leu 16 residues and the centre of mass of two whole Leu 45 residues from both protein chains. The second vector captured the orientation of the two short helices on the domain sides and was defined between the backbone N atoms of the Ile 7 residues. The third perpendicular vector was defined between the backbone N atoms of the Ile 31 residues and described the orientation of the two long helices. In simulations where the domain dissociated from the membrane, we did not evaluate the orientation of the domain with respect to the membrane.

## Figures and Tables

**Figure 1 ijms-22-08350-f001:**
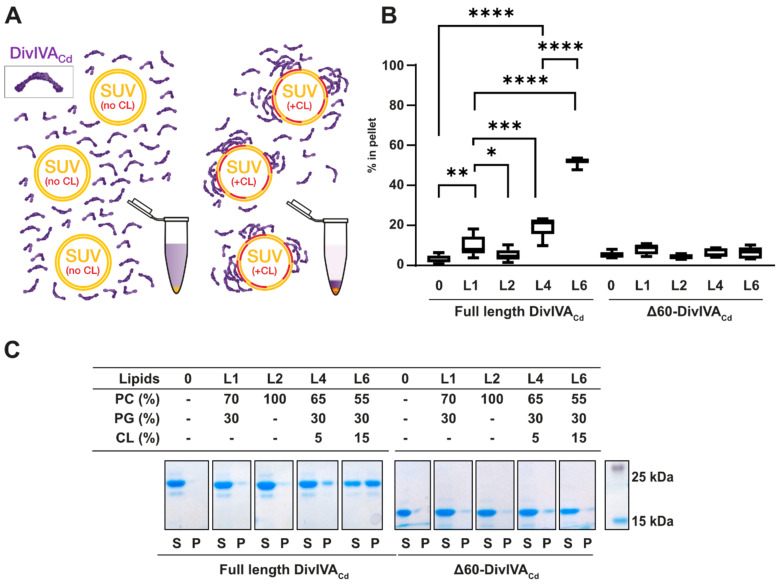
DivIVA_Cd_ binds better to SUVs containing CL and the N-terminal domain is essential for its binding. (**A**) A scheme of the sedimentation assay. DivIVA_Cd_ was mixed with solutions of SUVs that did not contain cardiolipin (in yellow) or contained SUVs with different cardiolipin concentrations (CL is in red). All SUV mixtures were prepared with the buffer containing 30% sucrose to enhance their sedimentation. The SUVs were pelleted by centrifugation and DivIVA_Cd_ amounts were quantified. The DivIVA_Cd_ illustration is based on the structure of the full-length DivIVA_Bs_ tetramer [38]. The sedimentation assay with N-terminally truncated Δ60-DivIVA_Cd_ was performed similarly. (**B**) Box and whisker plots of the percentages of DivIVA_Cd_ identified in the pellets under each of the tested conditions. Sedimentation assays were usually performed in triplets and in 3 independent experiments with full-length DivIVA_Cd_ samples. Sedimentation assays with Δ60-DivIVA_Cd_ were performed in triplets two times. Selected significant *p*-values from two-tailed unpaired *t*-tests are depicted above the plot; the threshold for statistical significance was taken to be *p* < 0.05. Stars indicate significantly different values, where * corresponds to *p*-value ≤ 0.05, ** to *p* ≤ 0.01, *** to *p* ≤ 0.001 and **** to *p* ≤ 0.0001. All *p*-values and percentages of DivIVA_Cd_ or Δ60-DivIVA_Cd_ in supernatants and pellets are listed in Appendix A. (**C**) An SDS-PAGE analysis of a representative sedimentation assay. The compositions of the individual SUVs tested are given above the gel. S refers to the soluble fraction; P, to the pellet.

**Figure 2 ijms-22-08350-f002:**
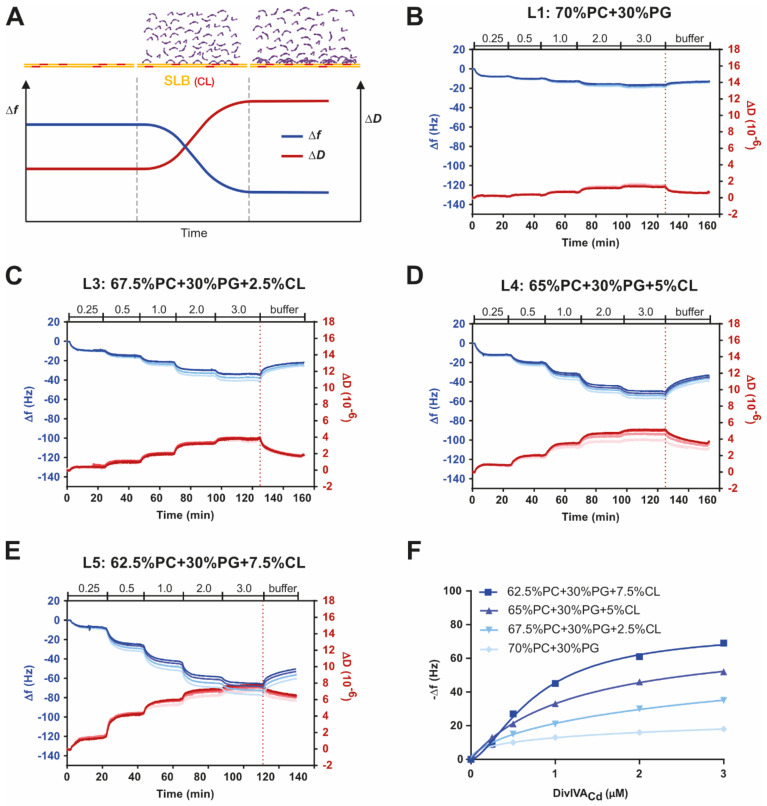
QCM-D measurements confirm that DivIVA_Cd_ binding depends on the lipid composition. (**A**) Simplified scheme of the QCM-D experiment. Full-length DivIVA_Cd_ (violet) was layered over the SLB on the QCM-D sensor chip. While the sedimentation assay investigated the binding of DivIVA_Cd_ to vesicles of different lipid composition, this experiment examined the ability of DivIVA_Cd_ to bind to different lipid bilayers. The DivIVA_Cd_ illustration is based on the structure of the full-length DivIVA_Bs_ tetramer [38]. The illustration was adapted from biolinscientific.com. (**B**–**E**) Results from measurements using supported lipid bilayers (SLBs) containing different cardiolipin concentrations. Layers were formed on a silicon dioxide-coated surface of a quartz crystal from a suspension of liposomes. DivIVA_Cd_ was added to the SLB stepwise, starting with 0.25 µM after *t* = 1 min, followed by 0.5, 1.0, 2.0, and 3.0 µM DivIVA_Cd_, as indicated on the top margin of each plot, and the frequency (Δf) and dissipation (ΔD) changes were measured. The plots show Δf (in blue) and ΔD (in red) of the third, fifth, seventh, and ninth overtones (lower overtones are in lighter shades, the highest overtone is the darkest shade) normalized to the baseline obtained after SLB formation. (**F**) A summary plot of the normalized Δf7 versus DivIVA_Cd_ concentration shows a clear preference of DivIVA_Cd_ for the SLB with CL. The Δf7 curve is reversed, meaning that the ascending curve indicates increased mass absorption. The series were fit non-linearly to the Hill equation.

**Figure 3 ijms-22-08350-f003:**
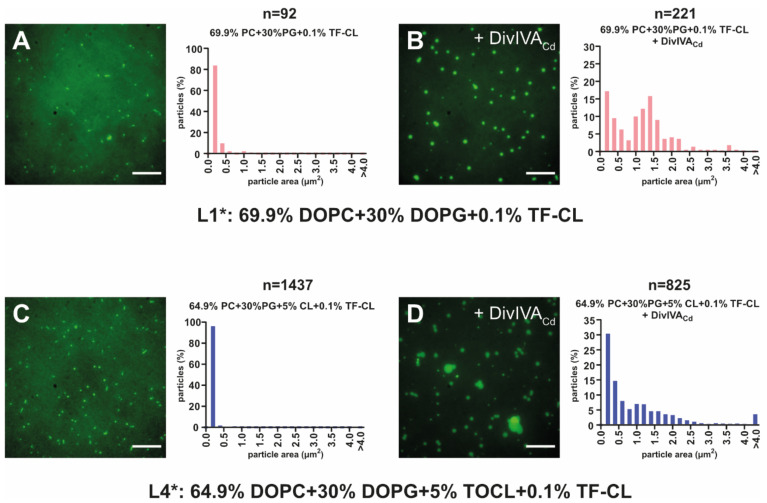
Changes to SLBs introduced by DivIVA_Cd_ visualized using fluorescently labelled cardiolipin. (**A**) Representative image of SLB composed of 69.9% PC, 30%, PG and 0.1% TF-CL (L1*, the asterisk designates the presence of fluorescent TopFluor^®^ Cardiolipin) before the addition of DivIVA_Cd_. Most of the fluorescent signal was homogenously distributed, with only few brighter foci of TF-CL visible. (**B**) The addition of 2 µM DivIVA_Cd_ to L1* caused changes to the fluorescence distribution, resulting in TF-CL clustering. However, we did not observe clusters with areas larger than 4 µm^2^. (**C**) SLB with 64.9% PC, 30% PG, 5% CL, and 0.1% TF-CL (L4*) before DivIVA_Cd_ addition. (**D**) Adding DivIVA_Cd_ to the chamber with an L4* SLB changed the lipid distribution and morphology of the bilayer even more visibly and larger clusters or foci can be observed (4% larger than 4 µm^2^). The scale bars represent 10 µm. The bar charts show the distribution of fluorescent areas from several images analysed using the built-in “Analyse Particle” plugin for automatic particle counting in Fiji. The number of particles (n) detected and measured is given at the top of each chart.

**Figure 4 ijms-22-08350-f004:**
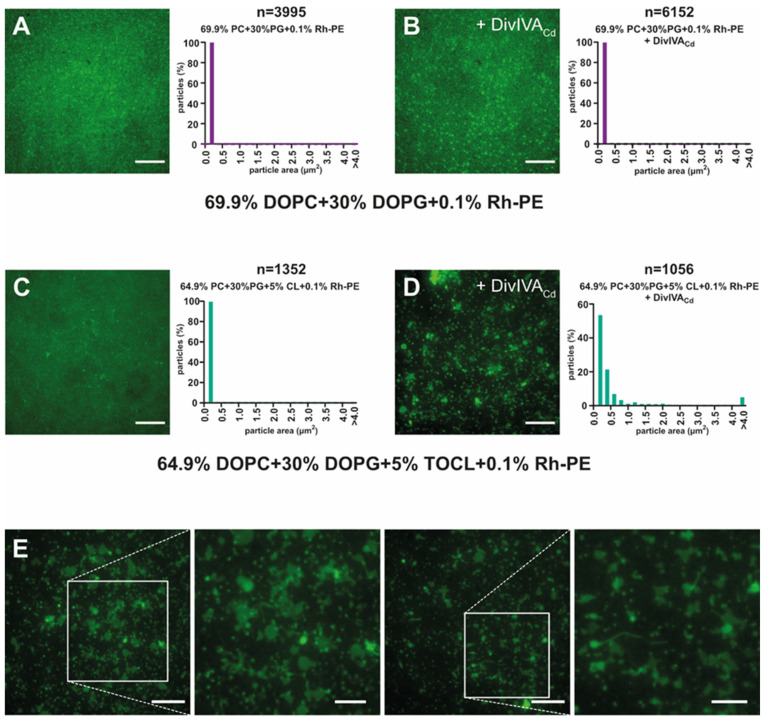
DivIVA_Cd_ binding also affects the lipid distribution in SLBs containing cardiolipin and labelled phosphatidylethanolamine. (**A**) The fluorescence in SLBs composed of 69.9% PC, 30% PG, and 0.1% Rh-PE (L1**) without DivIVA_Cd_ was evenly distributed (a double asterisk indicates the presence of fluorescent phosphatidylethanolamine in the lipid mixture designation). (**B**) In L1** samples, the fluorescence distribution remained unchanged after 2 µM DivIVA_Cd_ was added. (**C**) SLB containing 64.9% PC, 30% PG, 5% CL, and 0.1% Rh-PE (L4**) was also examined before the addition of DivIVA_Cd_. (**D**) The addition of 2 µM DivIVA_Cd_ to SLBs containing 5% CL caused changes in the lipid distribution and possibly morphological changes to the bilayer. Larger clusters formed, of which some measured several micrometres (5% larger than 4 µm^2^). (**E**) Details of the L4** bilayer changes observed in the presence of DivIVA_Cd_. The scale bars represent 10 μm in (**A**–**E**) and 5 μm in the magnified views.

**Figure 5 ijms-22-08350-f005:**
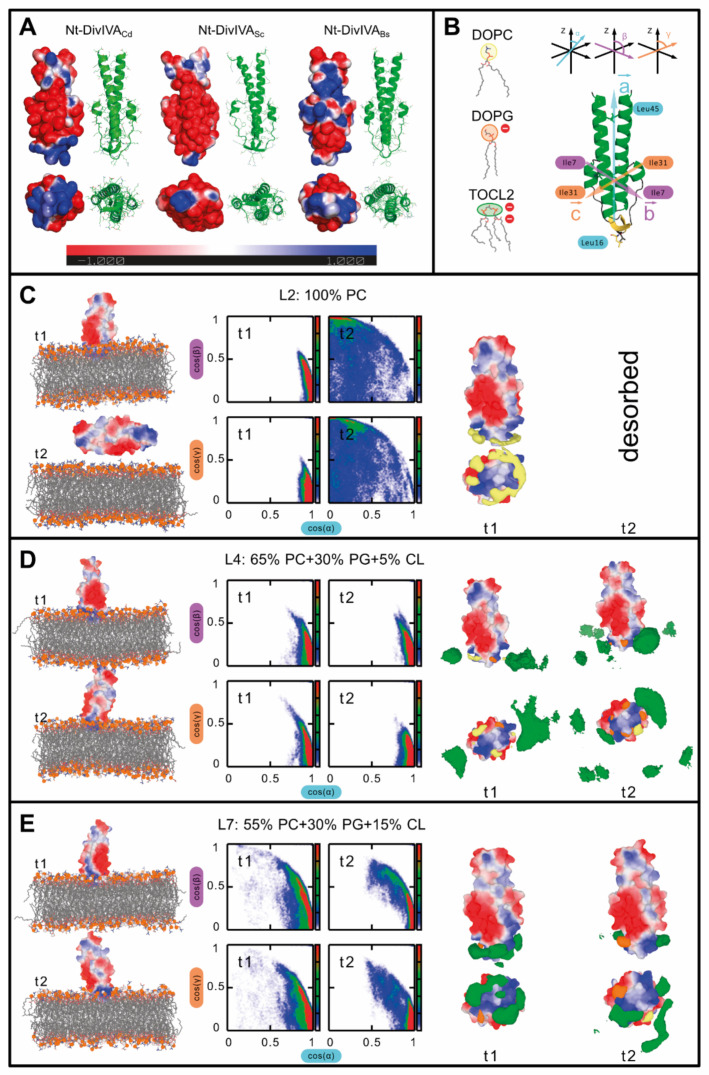
Simulations of the N-terminal domain of DivIVA_Cd_ binding to the membrane. (**A**) A comparison of the N-terminal domain models of *C. difficile* DivIVA_Cd_, *S. coelicolor* DivIVA_Sc_, and *B. subtilis* DivIVA_Bs_. The left view of each structure shows the electrostatic potential of the domain surface (the range is ±1 kT/e, negative is red, positive is blue). The right view shows the domain tertiary structure. The upper views show the side of the domain and the lower views show the view down the membrane-interacting end. (**B**) On the left are structures of the lipids used in the simulations. Coloured circles around lipid head-groups, with marked negative charges, are used to denote different lipid types in the density maps. On the right side is the DivIVA_Cd_ N-terminal domain with three orthogonal axes defined by vectors a, b, and c, which were used to describe the domain orientation with respect to the membrane. Angle α represents the tilt of the domain long axis, while angles β and γ capture the orientation of the domain sides. (**C**–**E**) Results from molecular dynamics simulations using 100% PC (L2), (**C**) 65% + 30% PG + 5% CL (L4), and (**D**) 55% + 30% PG + 15% CL (L6) bilayers. Each panel shows results from two independent simulation trajectories t1 and t2. Snapshots of the last frames of each simulation are shown on the left side; the molecular surface domain is shown coloured based on its electrostatic potential; the membrane lipids and phosphates are in grey and orange; water is omitted. The middle panels contain histograms showing the distribution of domain orientations with respect to the membrane, where cos(α) = 1 corresponds to a perpendicular orientation. The right-hand side shows the densities of the lipid head-groups around the domain tip.

**Figure 6 ijms-22-08350-f006:**
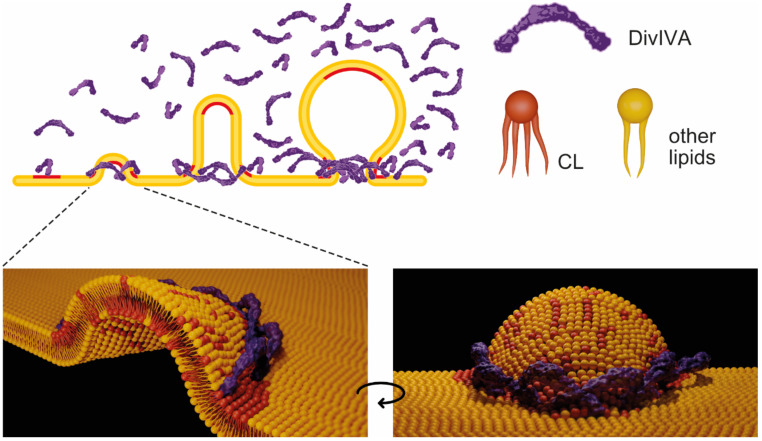
Model of DivIVA_Cd_ membrane binding. DivIVA_Cd_ preferentially binds to membrane areas with an increased cardiolipin concentration. Bound DivIVA_Cd_ attracts more DivIVA_Cd_ molecules and cardiolipin to these areas. The bilayer morphology is heavily distorted, and curvature develops. This, in turn, causes more cardiolipin to sort towards these areas and enhance DivIVA_Cd_ binding. The DivIVA_Cd_ illustration (violet) is based on the structure of the full-length DivIVA_Bs_ tetramer [38]. This illustration is simplified, however, as the protein forms higher order structures, and the shape and rigidity of these structures are likely to play a role in bilayer shaping. Cardiolipin (CL) is highlighted in red, other lipids are in yellow.

## Data Availability

The authors declare that the data supporting the findings of this study are available within the article and its Appendix A.

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
