# Peer review of "Cardiolipin-Containing Lipid Membranes Attract the Bacterial Cell Division Protein DivIVA"

_ijms, 2021, doi:10.3390/ijms22158350_

Round 1

Reviewer 1 Report

In this manuscript by Labajova, et al., the authors report that DivIVA from C. difficile binds preferentially to anionic phospholipids, especially cardiolipin, similar to S. coelicolor. Direct binding assays with composite vesicles are shown, as well as to supported lipid bilayers with additional imaging of lipid layers in TIRF. Additional clustering observations suggest that DivIVA can remodel the lipid bilayer through the formation of clustered regions.

1. Line 38: Change to “DivIVA is a late division protein”

2. Fig. 1A is unnecessary and does not add anything.

3. Fig. 1B is not referenced in the text.

4. Fig. 1- The typical bacterial lipid composition is approximately 70% PE, 20-30% PG and 5-10% CL. Only a small percentage of bacteria have PC. What is the composition in C. diff- does it have PC or PE? If it contains PE, one should use PE in the vesicle assays or try a commercially available purified E. coli lipid mixture.

5. Fig. 1- Was 55% PC/45% PG (or higher) tested? What about 100% PG versus 100% CL?. DivIVA may bind PG very well but may need a higher %. Cardiolipin properties precludes blilayer  formation, as in QCM-D, however it should be possible to construct SUVs with high CL %.

6. Line 224, do the authors mean “diphosphatidylglycerol”? Also, while the authors are correct in the CL has a -2 charge (and PG -1), CL also occupies twice as much physical space as PG, and 2 PGs could stack, mimicking a CL at the head group.

7. Does high ionic strength disrupt SUV and SLB binding similarly (also PG vs. CL)? Have more specific mutations in the positively charged region of DivIVA been constructed to test for loss of binding?

8. The clustering of TF-CL on the SLB is interesting. Does the clustering titrate with DivIVA concentrations over a range? Can the authors confirm that DivIVA colocalizes with the CL cluster. For example, could they use fluorescence-labeled DivIVA (i.e., Alexa fluor). Does DivIVA increase cluster size or decrease distance between clusters? Does cluster spacing change with PC/PG content?

9. Does the presence of DivIVA modify the FRAP half-time? Current 30 interval are too large to distinguish. One should retest with shorter intervals (i.e., 5s) and plot a time course.

10. Line 285- This reviewer is skeptical that the “tubular protrusions” described in Fig. 4E are real/relevant and not artifactual. Higher resolution techniques, such as electron microscopy with lipid vesicles, would provide more detailed insight.

11. Site directed mutagenesis of proposed Arg-rich regions would validate the underlying proposed biochemistry of the interaction (i.e., line 387).

Author Response

In this manuscript by Labajova, et al., the authors report that DivIVA from C. difficile binds preferentially to anionic phospholipids, especially cardiolipin, similar to S. coelicolor. Direct binding assays with composite vesicles are shown, as well as to supported lipid bilayers with additional imaging of lipid layers in TIRF. Additional clustering observations suggest that DivIVA can remodel the lipid bilayer through the formation of clustered regions.

We would like to thank the reviewer for all the comments to improve our manuscript.

  1. Line 38: Change to “DivIVA is a late division protein”

We have changed the text as suggested.

  1. Fig. 1A is unnecessary and does not add anything.

We would like to keep the scheme, as it makes easier for some of the readers, who are not familiar with this method to better understand the experiment.

  1. Fig. 1B is not referenced in the text.

Thank you, we have added the references for the Fig. 1B into the text.

  1. Fig. 1- The typical bacterial lipid composition is approximately 70% PE, 20-30% PG and 5-10% CL. Only a small percentage of bacteria have PC. What is the composition in C. diff- does it have PC or PE? If it contains PE, one should use PE in the vesicle assays or try a commercially available purified E. coli lipid mixture.

Indeed, C. difficile does not contain PC nor PE (Drucker et al., J. Bacteriol. 1996; Korachi et al., Anaerobe 2002; Guan et al., Biochim. Biophys. Acta 2014). We have chosen PC because of its overall neutral charge, cylindrical shape, and as is used routinely especially for SLB formation by the community. In our initial experiments, we were considering the usage of E. coli Total Lipid Extract or Polar Lipid Extract but beside PG and CL, both contain high amount of PE (57% and 67%, respectively), and the total lipid extract contains even 17% of unknown lipids. We therefore decided for more defined lipid compositions. Moreover, both mixtures are most possibly composed of lipid species with various acyl chains lengths, which can lead to formation of uneven SLBs, and also the conical shape of PE, could impose negative curvature stress on the SLB. As DivIVA is known to bind to negative curvature, we wanted to minimize the presence of initial curvature in our experiments to observe mainly effect of different lipid composition on DivIVA lipid binding.   

  1. Fig. 1- Was 55% PC/45% PG (or higher) tested? What about 100% PG versus 100% CL?. DivIVA may bind PG very well but may need a higher %. Cardiolipin properties precludes bilayer formation, as in QCM-D, however it should be possible to construct SUVs with high CL %.

Yes, we tested also higher concentrations of PG in sedimentation assays (e.g. 40%PC and 60%PG) and more DivIVA has bound to such SUVs. As suggested by other authors (Oliva et al., EMBO J 2010; Jurasek et al., Biochim. Biophys Acta - Biomembranes 2019) and as mentioned in the text possibly electrostatic interactions plays role in DivIVA binding, so it is not surprising. Importantly, the QCM-D and TIRF experiments point toward different effect of DivIVA binding on membranes with just 30%PG in comparison to binding to membranes with 30%PG + cardiolipin. Not only DivIVA binds better to cardiolipin-containing membranes, but also it causes changes in the viscoelastic properties, lipid distribution and possibly morphology of cardiolipin-containing membranes. Such changes were not observed with membranes without cardiolipin. Moreover, such high PG concentrations are non-physiological and because of already extensive length of the manuscript, we did not included those experiments.

We did not test 100% PG or 100% CL. However, we do not think this experiment would bring more light into our findings. Such conditions are completely artificial and can lead to unexpected changes in SUVs properties, e.g. instability or aggregation of SUVs, etc. and hence also contribute to changes in DivIVA membrane binding. Moreover, the sedimentation assay is not accurate method and the obtained results would not be possible to evaluate by additional methods like, QCM-D, as correctly mentioned by the reviewer.

  1. Line 224, do the authors mean “diphosphatidylglycerol”? Also, while the authors are correct in the CL has a -2 charge (and PG -1), CL also occupies twice as much physical space as PG, and 2 PGs could stack, mimicking a CL at the head group.

We thank the reviewer for pointing out the typo, we corrected it. 

We agree that electrostatically two PG molecules could mimic one CL molecules. However, chemical nature of both molecules is very different, especially at head group region. This difference is responsible for different propensity of both lipids to DivIVA that we observed in our simulations. Moreover, the changes to the membrane after DivIVA addition that observed in the presence of cardiolipin were not observed in membranes with even 30% PG. 

  1. Does high ionic strength disrupt SUV and SLB binding similarly (also PG vs. CL)? Have more specific mutations in the positively charged region of DivIVA been constructed to test for loss of binding?

We did not test high ionic strength on DivIVA binding. We did use maximally 250 mM KCl in our experiments with SUVs and DivIVA binding was not disrupted. In case of the second question, it is very likely that mutations in the N-terminal positively charged amino acids would interfere with DivIVA membrane binding, similarly as complete deletion of the N-terminal binding domain that we used in this work.

  1. The clustering of TF-CL on the SLB is interesting. Does the clustering titrate with DivIVA concentrations over a range? Can the authors confirm that DivIVA colocalizes with the CL cluster. For example, could they use fluorescence-labeled DivIVA (i.e., Alexa fluor). Does DivIVA increase cluster size or decrease distance between clusters? Does cluster spacing change with PC/PG content?

Thank you for this suggestion. We want to characterize this effect in more detail in the future work. We are currently preparing DivIVA-mScarlet fusion protein and fluorescently labelled DivIVA. However, we need to optimize the purification process as well as labelling of the protein. So far, we could not get a protein of sufficient quality to be involved in the experiments.

  1. Does the presence of DivIVA modify the FRAP half-time? Current 30 interval are too large to distinguish. One should retest with shorter intervals (i.e., 5s) and plot a time course.

We would like to point out that we used FRAP more as a control of quality of the bilayer and it was not used to test the effect of DivIVA binding. However, from available FRAP measurements both with CL and without CL the recovery half-times are comparable before and after DivIVA addition. We plan to perform more detailed FRAP analysis in the future to better characterize the effect of DivIVA on the membrane. Nevertheless, we would like to keep this FRAP analysis in the current manuscript, as an illustrative information in the supplementary material.

  1. Line 285- This reviewer is skeptical that the “tubular protrusions” described in Fig. 4E are real/relevant and not artifactual. Higher resolution techniques, such as electron microscopy with lipid vesicles, would provide more detailed insight.

Indeed, we also agree that the effect DivIVA binding has on the membrane morphology needs more detailed study. We are currently performing TEM and AFM studies with DivIVA and lipid vesicles and SLB, respectively. We changed the questionable statement in the manuscript concerning tubular protrusions.

  1. Site directed mutagenesis of proposed Arg-rich regions would validate the underlying proposed biochemistry of the interaction (i.e., line 387).

Thank you for this suggestion, we plan to prepare and test such mutants in the future.

Reviewer 2 Report

The research work reported in the present manuscript follows the groove marked out by the previous papers of this group on the bacterial DivIVA protein-lipids interactions.  The experimental design is well prepared and the results clearly support the hypotheses previously proposed, for example, in Biochimica et Biophysica Acta - Biomembranes 2019, 1862, 183144 as well as the new model proposed in this manuscript. The quality of the presentation is high and I recommend to the authors few changes to the text, which will improve its readability. 

-I suppose a problem occurred during the PDF construction since in the text are cited 89 references but the reference list ends at the 54th.

  • The figures are of good quality and interesting but the legends are definitely too long. Please, avoid repeating information already presents in the graphs or in the text. Reduce the size to the minimum necessary to explain the various components of each figure.
  • The text itself is quite verbose. Some obvious statements can be removed from Introduction, where you can mainly focus on your previous results and on how this work would enlarge the knowledge about this protein, and from the Results sections. In particular, the Discussion cannot be a summary of the paper, but it must concern just your comment on the outcomes of this research and the possible perspectives. 

Author Response

The research work reported in the present manuscript follows the groove marked out by the previous papers of this group on the bacterial DivIVA protein-lipids interactions.  The experimental design is well prepared and the results clearly support the hypotheses previously proposed, for example, in Biochimica et Biophysica Acta - Biomembranes 2019, 1862, 183144 as well as the new model proposed in this manuscript. The quality of the presentation is high and I recommend to the authors few changes to the text, which will improve its readability. 

We would like to thank the reviewer for all the comments to improve our manuscript.

-I suppose a problem occurred during the PDF construction since in the text are cited 89 references but the reference list ends at the 54th.

We have added the missing references. In addition, because of text reductions, we changed the whole list of references accordingly.

  • The figures are of good quality and interesting but the legends are definitely too long. Please, avoid repeating information already presents in the graphs or in the text. Reduce the size to the minimum necessary to explain the various components of each figure.

We have reduced the most extensive legends of Fig.1 and 2.

  • The text itself is quite verbose. Some obvious statements can be removed from Introduction, where you can mainly focus on your previous results and on how this work would enlarge the knowledge about this protein, and from the Results sections. In particular, the Discussion cannot be a summary of the paper, but it must concern just your comment on the outcomes of this research and the possible perspectives. 

As suggested we have reduced the text and changed especially Introduction and Discussion parts. However, we would like to keep simplified summary of results in Discussion as there are in context with in vivo observations from the field and we proposed a new model which should be further tested.  

Round 2

Reviewer 1 Report

The authors have satisfactorily addressed my immediate concerns. Additional experiments will be insightful in the future for a more complete understanding of the nature and biological relevance of the observed interaction.